# Behavioral Inhibition and Social Competence Through the Eyes of Parent and Teacher Informants

**DOI:** 10.3390/bs14111080

**Published:** 2024-11-12

**Authors:** Hailey Fleece, Hedwig Teglasi

**Affiliations:** Department of Counseling, Higher Education, and Special Education, University of Maryland, College Park, MD 20742, USA; hfleece@umd.edu

**Keywords:** behavioral inhibition, temperament, social competence, informant discrepancy, early childhood, person perception, goodness-of-fit

## Abstract

The centrality of social competence to children’s well-being has sparked interest in documenting its correlates and precursors. Behavioral Inhibition (BI) is studied extensively as an early appearing, biologically based, temperamental disposition that places children at increased risk for maladaptive social functioning. Children with BI are characterized by the tendency to react to unfamiliarity or uncertainty with fear and to respond with avoidance or withdrawal, eventuating in missed opportunities to gain social competence (SC). Early interventions that aim to interrupt this negative cycle often rely on parents or teachers to observe BI, but they often disagree in their ratings, raising understudied but basic questions about how to translate the research findings into effective interventions. In this study, parents and teachers rated kindergarteners’ (N = 174) disposition toward fear and shyness, underpinnings of BI and SC. As expected, we found modest overlap in the classification of children into relatively High, Average, and Low BI groups based on parent and teacher ratings. Whereas about 40 percent were classified similarly, about 33 percent were discrepant in their classification by more than one category. Yet, the High BI group was at a social disadvantage (lower SC) compared to the Low BI group, even when the comparison groups only included children whose classification was discrepant. In line with the Realistic Accuracy Model of person perception, we describe a context/informant-specific conceptualization of the BI–SC connection with implications for intervention.

## 1. Introduction

The centrality of social functioning to young children’s short- and long-term well-being [1] underscores the importance of understanding its precursors and correlates. In Western cultures, a large body of work shows that Behavioral Inhibition (BI) places children at a social disadvantage [2]. BI refers to a biologically based tendency to respond to unfamiliar people, objects, or situations with fear, restraint, or withdrawal that is evident from the first year of life and associated with increased risk for the later emergence of interpersonal difficulties and anxiety disorders [3,4]. Early BI predicts relatively ineffective social problem-solving with unfamiliar peers during the toddler and preschool years [5] and continues to impact the life course, predicting multiple outcomes, including less effective social functioning with family and friends during the adult years [6].

A sizable proportion of children with an early history of BI subsequently experience problems with anxiety (40%), establishing the prominence of BI as a risk factor [3] that has sparked considerable interest in the processes, within and outside the child, by which BI does or does not eventuate in a mental health disorder [7]. According to the developmental cascade model, the impact of temperamental traits on outcomes is magnified over time by influencing the individual’s functioning in other domains, such as social competencies [8] that play key roles in well-being [9,10]. In a sense, the relation of BI with lower SC is a proximal indicator of a negative cascade because less effective social behavior disrupts the goodness-of-fit, which refers to a match between the child’s temperament and the functional requirements of the context [11,12]. In turn, a poor fit is a potential avenue by which the adverse effects of BI are further magnified.

Concurrent associations of BI with SC are snapshots of a process by which the adverse impact of BI on adjustment may be magnified over time. As children enter formal schooling, some children may experience distressing emotions underlying BI, such as fear and shyness, more frequently and intensely than do others and in response to a wider range of interpersonal contexts, and these emotions interfere with socially effective behavior [13,14]. In turn, less effective social behavior places children at a disadvantage compared to peers. For example, in a community sample, children with lower SC in first grade were less well-adjusted in third grade [15]. The adverse effects of BI are thought to occur through a bidirectional pathway where the child is biologically predisposed to withdraw from the peer group, which then limits the development of social competence (SC), leading to further withdrawal that sometimes eventuates in exclusion by peers [2]. From this standpoint, less effective SC, though related to BI, takes on a life of its own to increase risk for maladaptive exchanges over time.

Parents and teachers, natural allies in identifying and helping children to manage distressing emotions and navigate social challenges, are frequently called on to provide information about the emotions and behaviors of young children, including their BI reactivity and social functioning [16]. Yet, parent and teacher informants often differ when rating the same child’s BI tendencies [17] and social functioning [18]. They even disagree about the behaviors, emotions, and skills of children with BI [19]. Given the ubiquity of discrepancies, practitioners and researchers are advised to obtain information from multiple informants [20], but frameworks for understanding inconsistencies among informants are vastly understudied. Yet, the reality of informant discrepancies raises questions about how to conceptualize empirical relations among theoretically linked variables. Focusing on BI and its relation to SC, we propose that the Realistic Accuracy Model (RAM) of person perception accounts for informant discrepancies in rating each and argues for a context/informant-specific perspective on the relations between them. The RAM posits that the more important a trait is for functioning (trait relevance) in each setting, the more likely it is to be expressed and, hence, to be recalled and reported by informants. Discrepancies arise when informants are situated in settings that vary in the extent to which the trait in question bears on functioning. Parents and teachers would not be expected to agree unless the trait is similarly relevant in their respective settings (functional equivalence). Hence, informants may differ in their ratings of a trait but the relation of the rated trait to functioning may be similar across informants. From the RAM perspective, the finding that parents and teachers view different levels of a given behavior as warranting treatment [21] may also be explained by differences in trait relevance. For example, internalizing behaviors are likely of concern to parents and teachers, but may appear less noticeable, hence less salient, to teachers than to parents [22], most likely because internalizing behaviors generally do not disturb class lessons [23].

Studies vary in relative emphasis in conceptualizing BI as a stable temperamental trait or as a contributor to children’s bidirectional transactions with the surroundings, with attendant implications for explaining informant discrepancies. As a trait, BI is defined as a biologically rooted disposition that is relatively stable, situationally consistent, evident early on, and persisting beyond childhood [24]. Emphasis on BI as a stable trait associated with increased risk for adverse developmental outcomes does not readily accommodate informant discrepancies, apart from methodological flaws or random error. On the other hand, a transactional perspective, which emphasizes the role of BI in shaping how the individual responds to the surroundings and how others respond in turn [2], easily accommodates informant discrepancies, particularly if observers are situated in different contexts, such as parents and teachers. For example, although inhibited preschoolers are less likely than most peers to initiate or engage in social play behaviors [25], their reluctance to participate is exacerbated when they are experiencing uncertainty about what to expect or about how to respond, such as in novel or ambiguous situations [26].

### 1.1. Explaining Informant Discrepancies from a Transactional Perspective

There is broad consensus that informant discrepancies cannot be dismissed as measurement error [27], but instead, should be embraced as capturing the unique perspective of each informant [21,28]. Although many factors within the child and the informant contribute to informant discrepancies, recent work highlights the role of context [29]. The Realistic Accuracy Model (RAM) [30] provides a compelling framework to explain discrepancies between parent and teacher informants and offers insights about why behaviorally inhibited children may act or be perceived differently by informants situated in different environments.

The RAM’s suppositions regarding trait relevance and functional equivalence map onto the transactional perspective on children’s temperament. BI would be considered relevant for functioning within a context to the extent that it influences how the individual navigates the expectations of that context. The more BI interferes with functioning, given the conditions of the context (e.g., requirements, guidance, support, consistency), the more likely it is to be observed and reported by informants and the more likely it is to be associated with key aspects of functioning, including SC. Accordingly, an informant is likely to attend to a child’s tendency toward BI reactivity if it is relevant to functioning in the observed setting, regardless of the child’s functioning in other settings. Notably, the RAM framework implies that associations between a trait, such as BI, and SC, a proxy for functioning, is meaningful from the perspective of each informant, regardless of the other informant’s rating [18].

With respect to the relation of BI to social functioning, it is important to keep in mind that SC is not a unitary trait but a multi-faceted construct encompassing specific skills that enable effective responses in line with social expectations and individualistic social–emotional goals [31]. The relevance of specific aspects of SC for effective functioning (i.e., Communication, Cooperation, Assertion, Responsibility, Empathy, Engagement, and Self-Control [32]) depends on the requirements of the setting. Parents and teachers tend to focus on different aspects of SC [29] and they do not necessarily agree on which skills are most important for young children to develop. One study, for example, reported that parents and teachers agreed on only four of ten skills as the most relevant for child functioning [33]. For children with relatively high BI, some skills may be challenging in multiple settings (e.g., assertion, communication, or active engagement) and others (e.g., empathy, responsibility) may depend on the conditions in the context.

### 1.2. Current Study

Research has established concurrent and predictive associations between reactive tendencies underlying BI and children’s social functioning [14,34,35]. Kagan and colleagues [36] estimate that BI characterizes between 10 and 15 percent of young children, but the temperamental traits underlying BI, fear and shyness, are distributed along a normal spectrum. Emotional distress, underlying BI, including fear and shyness, appears to place young children at a social disadvantage [34,37]. Given the ubiquity of informant discrepancies [18], we anticipate low agreement between parent and teacher informants on the relative standing of individual children on BI, which raises questions about how to conceptualize its documented associations with SC. Based on the transactional view of BI and the explanation offered by the RAM for informant discrepancies, we argue for shifting to a context/informant-specific lens for conceptualizing the relative social disadvantage associated with higher BI.

The RAM framework for explaining informant discrepancies suggests that the association between a trait and its functionally linked correlates (i.e., BI and SC) would be specific to the informant observing in each setting. Correlational studies showing inverse associations between BI and SC may mask the role of context or informant.

In this study, grouping children separately according to relative standing on BI based on parent or teacher ratings allowed us to test hypotheses to unpack its documented links with SC. We formulated three hypotheses, consistent with our assumption that associations of BI with SC capture the child’s transactions as observed by each informant, regardless of the other.

**Hypothesis 1a.** 
*Considering that informant discrepancies are ever-present and that correlations across informants are weak or non-significant, we hypothesized that about half the children or fewer would be similarly classified into respective tercile groups (High, Average, and Low BI).*


**Hypothesis 1b.** 
*We expected similarly low overlaps even when the top and bottom 15% were designated as the High BI and Low BI groups, respectively. As a risk factor, BI is considered High beyond one standard deviation in relative standing. Support for this hypothesis suggests the importance of informant/context-specific approaches to early identification.*


**Hypothesis 2.** *We hypothesized that kindergarteners who are classified from the perspective of a single informant into the upper third, with respect to BI, would be rated lower on SC when compared to children classified in the lower third. We did not pose hypotheses about specific skills, but generally expected that patterns of BI group differences within informants would reflect the relevance of BI for skills that are valued in the observed setting*.

**Hypothesis 3.** *From a context/informant-specific perspective, it is reasonable to expect that the relative social disadvantage of children in the High BI group would be evident within each informant perspective, even when the groups comprised children whose relative standing on BI was rated differently by the other informant (children with similar classification were excluded). Support for this hypothesis would argue that BI and its relation to functioning should be identified and addressed in the setting(s) in which the concern arises, even if other informants disagree. For example, a child rated with higher BI, as observed by the teacher but not by the parent, would still be at a social disadvantage in that classroom setting*.

## 2. Materials and Methods

### 2.1. Participants

Participants included 149 parents and 37 teachers of 174 kindergarten students. A total of 125 children had complete data from a parent and a teacher; 131 children had data from a parent, while 158 children had data from a teacher. The students ranged in age from 4 to 6 years old (M = 68.70 months, SD = 4.96 months). Parents identified their child’s sex as male (56.30%) or female (43.70%). Participants identified their race and or ethnicity as follows: 60.24% White, 12.65% Asian, 10.84% Black, 8.43% Latinx, and 7.83% identifying as a different race or as multi-racial. Based on reported occupations, most children came from middle-class families. Students and teachers came from 10 different schools, 9 in the D.C. Metro area and 1 in the Chicago area. The teachers and students were primarily from independent schools or a laboratory research school at a public university. Each teacher reported on an average of 4.76 children (SD = 4.77). All teachers identified as White and female. Differences between schools accounted for two percent of the variance (F = 0.328, *p* = 0.57) and differences between teachers accounted for three percent of the variance (F = 0.408, *p* = 0.52) in teacher-reported BI.

### 2.2. Measures

#### 2.2.1. Social Competence (SC)

The Social Skills Improvement System-Rating Scales (SSIS-RS) [38] is designed to be completed by parents and teachers to evaluate children’s social behaviors in the home and school settings and identify significant social skills deficits. The SSIS includes 140 items, divided into three broad sections: Social Skills (“Makes friends easily”), Problem Behaviors (“Withdraws from others”), and Academic Competence (“Academic performance compared to group”; teacher form only). Informants rate how often they have seen the child engage in certain behaviors over the past two months on a 4-point scale (0 = Never, 1 = Seldom, 2 = Often, and 3 = Almost Always). A composite Social Skills score is obtained by averaging all items in the Social Skills subscales. Given the goals of this study, we analyzed the Total SSIS score, seven subscales measuring specific skills (Communication, Cooperation, Assertion, Responsibility, Empathy, Engagement, and Self-Control), and the Internalizing domain of the Problem Behaviors Scale. Concurrent validity has been established with other measures of social, emotional, and behavioral functioning in children [38]. In the current sample, internal consistency for the subscales is high, ranging from 0.77 to 0.97 for both parent and teacher forms (median 0.90 for teachers; 0.85 for parents).

#### 2.2.2. Behavioral Inhibition (BI)

The Child Behavior Questionnaire (CBQ) [39] provides a detailed assessment of temperament in young children. In this study, parents completed the Short Form of the CBQ [40] and teachers completed the adaptation of the Short Form for teachers (CBQ-TSF) [41]. Both measures contain 94 items that capture the expression of 15 childhood temperamental characteristics using seven-point Likert scales ranging from “extremely untrue of the child” to “extremely true of the child”. To obtain a measure of BI as defined by Kagan and colleagues [36], we averaged scores of two CBQ scales, Shyness and Fear, separately for parent and teacher informants. In this study, the Shyness and Fear subscales were weakly correlated, though significant, for both teachers (r= 0.302, *p* < 0.001) and parents (r = 0.278, *p* < 0.001), suggesting that despite similarities in the underlying constructs, each variable is tapping into unique aspects of BI. The CBQ-SF shows strong correlations when completed by the same rater 2 to 3 years later (r = 0.50–0.79 [40]). In the current sample, internal consistency was acceptable for both the fear (T α = 0.70, *p* α = 0.74) and shyness (T α = 0.88; *p* α = 0.86) subscales.

We categorized children into three equal-sized groups based on relatively High, Average, or Low BI scores, split at the 33rd and 66th percentile, separately for parent and teacher informants. In dividing the continuous BI reactivity variable into three equal categorical variables, we emphasize the idea that variations between the three groups reflect meaningful differences in the impacts of BI on social functioning. It is not the presence of BI but the relative standing on the trait that places a child at risk of social difficulties [42].

To mitigate the risks of grouping a continuous variable, a sensitivity analysis was run. Linear regression was used to examine the relationship between BI and social skills as reported by parents and teachers. Apart from one variable (teacher-reported Cooperation), the results were consistent between the two analyses used. Given the consistency of the findings, categorization was utilized because it allows for parents, teachers, and practitioners to take meaningful implications from the study.

### 2.3. Procedure

In accordance with the IRB protocol, informed consent and questionnaires were distributed to parents of kindergarteners in the 10 participating schools. With parent consent, the student’s primary teacher also filled out the questionnaires. To ensure that the teacher had time to get to know the student, data collection occurred after October. Graduate researchers reviewed completed scales and followed up with parents and teachers who skipped items to minimize missing data.

## 3. Results

### 3.1. Descriptive Information

Some parents and teachers selected the Not Applicable (N/A) option or missed an item in their rating. On the CBQ, 0.7% of all parent items and 6.3% of all teacher items were marked N/A. Only 0.08% of parent items and 0.07% of teacher items were left blank.

Guidelines for the CBQ were followed, which indicate that missing items should be replaced by the mean of the available responses for the items of the scale. Teacher ratings were available for 153 children and parent ratings were available for 129 children. A paired-samples *t*-test comparing BI reactivity as reported by parents and teachers showed a significant difference by reporter (t(131) = 10.50, *p* < 0.001). On average, parents (M = 3.77, SD = 1.09) reported significantly higher BI reactivity scores than did teachers (M = 2.65, SD = 0.88) and teachers reported higher Overall Social Skills (t(137) = 4.03, *p* < 0.001) than did parents. Means and standard deviations for social competency scores based on BI reactivity classification for both parents and teachers for the full and subsamples are displayed in Table 1 and Table 2, respectively.

#### 3.1.1. Preliminary Correlational Analyses

In the following correlational analyses, BI was treated as a continuous variable, based on an aggregate of fear and shyness. 

##### Informant Agreement

The bivariate Pearson’s correlation between parent- and teacher-rated BI reactivity was modest, though significant (r = 0.24, *p* < 0.001). Likewise, the correlation between parent- and teacher-rated Total SC scores was modest, though significant (r = 0.23, *p* < 0.001). Similarly weak associations were found across informants on specific social skills (correlations ranged from r = 0.14 to r = 0.24).

#### 3.1.2. Associations Between BI and SC

##### Within-Teacher Report

Teacher-rated BI reactivity correlated with the following SSIS scales: Assertion (r = −0.22, *p* = 0.01), Engagement (r = −0.35, *p* < 0.001), and Internalizing behaviors (r = −0.40, *p* < 0.001) but not with other scales (Communication, Cooperation, Empathy, Responsibility, Self-Control, or Overall Social Skills).

##### Within-Parent Report

Parent-rated BI reactivity correlated with most social competencies, including Assertion (r = −0.37, *p* < 0.001), Communication (r = −0.30, *p* < 0.001), Empathy (r = −0.18, *p* = 0.04), Engagement (r = −0.56, *p* < 0.001), Self-Control (r = −0.23, *p* = 0.01), Internalizing Problems (r = −0.34, *p* < 0.001), and Total Social Skills (r = −0.40 *p* < 0.001), but not with Responsibility and Cooperation. Within both parent and teacher reports, Higher BI was associated with lower Engagement, lower Assertion, and higher Internalizing. Parent-rated BI correlated with almost all SSIS scales (six of eight) and with Internalizing Problems.

##### Between-Informant Report

Parent-reported BI reactivity did not correlate with any teacher-reported SC scales, whereas teacher-reported BI reactivity correlated with three parent-reported SSIS scales, though modestly: Cooperation (r = 0.18, *p* = 0.04), Responsibility (r =0.18 *p* < 0.001), and Engagement (r = −0.24 *p* < 0.001), as well as with Internalizing Problems (r = −0.30, *p* < 0.001). These weak cross-informant correlations speak to the context/informant specificity of the relations between BI and SC.

### 3.2. Informant Overlap in the Classification of Children

Findings supported the expectation that fewer than half of the children would be similarly classified by both parents and teachers.

**Hypothesis 1a.** 
*Classification into terciles.*


Overall, 53 of the 130 children (40.77% of the sample) were similarly classified based on both parent- and teacher-rated BI (Low, Average, or High BI reactivity). In the Low BI group, parents and teachers agreed on 18 children (Parent: M = 2.50; SD = 0.66. Teacher: M = 1.60; SD = 0.50). In the Average BI group, informants agreed on 16 children (Parent: M = 3.80; SD = 0.28. Teacher: M = 2.63; SD = 0.18). In the High BI group, informants agreed on 19 children (Parent: M = 5.07; SD = 0.46. Teacher: M = 3.75; SD = 0.81). About one in three children were rated differently by more than one categorical level (e.g., rated Low BI by teacher and High by parent).

**Hypothesis 1b.** 
*Classification into upper and lower 15%.*


The pattern described above held even when looking at the most and least inhibited children. An examination of the top 15% of inhibited children revealed that parents and teachers only agree on their ratings about half the time. Parents agreed on the high BI rating for eleven of the nineteen children with the highest teacher-reported inhibition. Teachers agreed on the high BI rating for nine of the nineteen children with the highest parent-reported inhibition. Parents agreed on the low BI rating for eleven of the nineteen children with the lowest teacher-reported inhibition. Teachers agreed on the low BI rating for seven of the nineteen children with the lowest parent-reported inhibition.

### 3.3. Hypothesis 2: Comparing Informant-Specific BI Groupings on SC

A series of one-way ANOVAs was conducted to compare the effects of BI reactivity on social competencies as reported separately by parents and teachers (Table 3). Findings supported the hypothesis that, from the perspective of each rater, the children classified in the High group would be at a social disadvantage relative to children classified in the Low group.

#### 3.3.1. Teachers

In keeping with the hypothesis, there was a significant effect of BI reactivity on Assertion, Cooperation, Engagement, and Internalizing Problems. There was no significant effect on Communication, Empathy, Responsibility, Self-Control, or Total Social Skills. Post hoc comparisons using the Tukey HSD test indicated that children in the relatively High teacher-reported BI reactivity group have significantly lower teacher-reported Assertion (*p* = 0.03) and Engagement (*p* < 0.001), and higher Internalizing Problems (*p* < 0.001) than their peers in the Low BI reactivity group. Children with Average levels BI reactivity have significantly higher levels of Cooperation than their peers with Low BI (*p* = 0.03). As shown in Figure 1, most of the differences occurred between High and Low BI groups, except for Cooperation, where the Average BI group had the highest score.

#### 3.3.2. Parents

In keeping with the hypothesis, for parents, there was a significant effect of BI reactivity level on Assertion, Communication, Engagement, Self-Control, overall Social Skills, and Internalizing Problems. No significant effects were found for parent-reported Cooperation, Empathy, or Responsibility. Post hoc Tukey HSD analyses revealed that children in the relatively High parent-reported BI reactivity group have significantly lower Assertion (*p* < 0.001), Communication (*p* < 0.001), Engagement (*p* < 0.001), Self-Control (*p* = 0.03), and overall Social Skills (*p* < 0.001), and higher Internalizing Problems (*p* < 0.001) when compared to peers in the relatively Low BI reactivity group. Findings are depicted in Figure 2.

### 3.4. Hypothesis 3: Comparing BI Groupings When Agreed-Upon Children Were Excluded

Despite the reduced number of participants in the analyses, findings supported the hypothesis that the relative social disadvantage of children in the High BI group compared to the Low BI group would be evident within each informant’s perspective, even when the other informant disagreed.

One-way ANOVAs were conducted to examine within-informant BI group differences on SC in a subsample of children whose parents and teachers did not agree on BI reactivity classification (n = 74; Table 4).

#### 3.4.1. Teachers (Subsample)

For teachers, a significant effect of BI reactivity remained on Engagement, and Internalizing Problems, with the effect on Assertion no longer reaching significance. Children in the High BI reactivity group had lower teacher-reported Engagement (*p* < 0.001) and higher Internalizing Problems (*p* < 0.001) when compared to children in the Low BI reactivity group.

#### 3.4.2. Parents (Subsample)

For parents, a significant effect of BI reactivity remained on Assertion, Communication, Engagement, and Total Social Skills, though the effect of BI on Self-Control and Internalizing did not reach significance. Tukey HSD post hoc comparisons indicated that children in the High BI group have significantly lower parent-reported Assertion (*p* = 0.01), Communication (*p* < 0.001), Engagement (*p* < 0.001), and Total Social Skills (*p* < 0.001) than their peers in the Low BI group.

### 3.5. Sensitivity Analysis

A sensitivity analysis using simple linear regression was conducted to examine the relationship between BI as a continuous variable and social skills as reported by parents and teachers. Results were consistent with the ANOVAs except for one variable, teacher-reported Cooperation. BI reactivity as a continuous variable did not predict teacher-reported Cooperation (F(2,158) = 3.46, *p* = 0.06). However, when grouped into three categories, BI showed a nonlinear relationship with teacher-reported Cooperation. Teachers rated children with Average BI as highest in Cooperation with significantly higher scores than their Low BI peers (this trait did not vary across parent-rated BI groups). Teachers regard inhibitory control of behavior as a very important trait [33,43] and a moderate degree of fear and shyness appears to be relevant to cooperation in the classroom. The nonlinear nature of the relationships between cooperation, a key social skill, and BI, along with informant differences, highlight the benefits of grouping children according to relative BI ratings.

## 4. Discussion

Theoretical models of development implicate temperament as a source of individuality with direct and indirect influences on children’s adjustment, ranging from social functioning to behavioral symptoms, such as internalizing problems [39]. Behavioral Inhibition (BI) is defined in terms of a set of observable behaviors, including social wariness and withdrawal, that are presumably motivated by underlying temperamental tendencies toward fearful reactivity and shyness [7]. A substantial body of work shows that children with higher BI tend to be less socially effective than their peers [34,37]. As children enter formal schooling, concerns about their social or academic adjustment are flagged by parents or teachers, but they often disagree. Each informant is thought to provide unique insights, warranting consideration of multiple perspectives in both practice and research settings [29]. The problem is that the lack of empirically supported explanations of informant discrepancies [18] limits the quality of services for children and muddies psychological theory.

The inverse relation of BI with social functioning has garnered theoretical and empirical support, but the translation of this research into practice is incomplete without consideration of the issues raised by informant discrepancies. Patterns emerging in this study are consistent with the Realistic Accuracy Model of person perception (RAM) [30] as a framework for understanding the factors contributing to informant ratings and of informant discrepancies. The RAM posits that informants are more likely to notice and report traits that are more relevant to aspects of functioning that are expected or valued by the observer. Hence, higher mean parent- than teacher-rated BI reactivity, found in this study, would suggest that parents regarded BI as more relevant to the child’s functioning than did teachers, and higher teacher- than parent-reported SC, found in this study, would suggest that teachers regarded such skills as more relevant than did parents. Prior research shows that teachers place more emphasis on social skills for their students than do parents [33]. The relevance of BI reactivity for a particular individual may vary with the external context (situation) and the internal (other attributes) context, and these differences in relevance for functioning in the observed setting are evident in the weak associations found in this study between parent- and teacher-rated BI (0.24) and SC (0.23). These informant discrepancies in rating each attribute have implications for understanding the relations between them. If, as argued by the RAM, an informant is more likely to report BI if it is associated with social functioning in that setting, for which SC is a proxy, then the inverse association between them is to be expected, but only if both attributes are rated by the same informant.

In this study, significant inverse correlations emerged between BI and SC, but primarily within informants. Weak or non-significant cross-informant relations between BI and SC are consistent with the context/informant specificity of the ratings themselves, which are thought to capture the transactions in each setting, as observed by the informant. We note, however, that cross-informant BI–SC relations varied somewhat depending on the informant rating of BI. Whereas parent-reported BI did not correlate with any teacher-reported SC subscales, teacher-reported BI correlated modestly with three parent-reported subscales, Cooperation (r = 0.18, *p* = 0.04), Responsibility (r = 0.18, *p* < 0.001), and Engagement (r = −0.24, *p* < 0.001), as well as with Internalizing Problems (r = −0.30, *p* < 0.001). Although additional research is needed to explain this informant-specific pattern in BI–SC relations, one may speculate that teacher-rated BI may be more in line with normative considerations, hence may be somewhat more generalizable to certain aspects of functioning, particularly Internalizing Problems, in other contexts.

To delve further into the informant-specific BI–SC correlations, we compared SC scores of children classified into relatively High, Average, and Low BI groups based on parent and teacher ratings. In accord with hypothesis 1a and consistent with low cross-informant correlations on BI ratings, fewer than half (about 40%) were similarly classified into the High, Average, and Low BI tercile grouping based on both parent and teacher ratings. Of the children who were classified into different groups, one third were rated differently by more than one categorical level (e.g., Low BI according to one informant and High BI according to the other). Given that concerns are usually noted as clinically significant when departure from the norm is more extreme, we also examined overlaps between groups at the top and bottom 15th percentile. In accord with hypothesis 1b, we found similarly low overlaps when classifying the most and least inhibited children (52 and 47 percent overlap, respectively). These findings are consistent with research showing that associations among informants’ ratings of problematic behaviors depend more on the informant pairs than on the nature of the problem [44].

**Hypothesis 2.** 
*We compared BI groups on aspects of SC in two ways, one that included all participants and one that excluded children who were similarly classified into BI groups based on the other informant’s rating (i.e., discrepant groupings). As hypothesized, from the perspective of a single informant, children rated in the upper tercile on BI reactivity were rated lower on aspects of SC than those in the lower tercile. This pattern held within both parent and teacher informants, even if the comparison groups comprised children with discrepant classifications. This means that in each setting, an informant who perceives a child as relatively high in BI also perceives that child as relatively low in SC, regardless of how the other informant views the child. Although parents and teachers perceive children differently with respect to their relative standing on BI, the relation of BI with SC remains.*


Informant discrepancies are commonplace, and explanations are important because they influence educational and psychological practice and research, including how BI is conceptualized and identified as an early appearing risk factor that warrants intervention. The emergence of BI group differences in SC when children with agreed-upon classification were excluded from analyses underscores the value of an informant-specific, transactional lens on the relation of BI with SC. The transactional perspective fits current approaches to temperament-based interventions that aim to ameliorate the adverse effect of BI by improving the ‘goodness-of-fit’ between the child and the environment [8,11]. Parents and teachers are well-positioned to detect BI and to respond in ways that mitigate its detrimental impact on social functioning within their respective settings [45]. Considering the potential benefits of home–school collaboration for children’s educational and psychological development, the National Association of School Psychologists 2005 (NASP) published a position statement on the importance of “families and educators working together to develop shared goals and plans that support the success of all students” [46] (p. 1). Home–school partnerships may be enhanced by empirically supported explanations of differences between parent and teacher perceptions.

**Hypothesis 3.** 
*Patterns of distinctions and commonalities across informants in the relations of BI with certain aspects of SC shed light on the relevance of BI traits for functioning vis-à-vis those competencies in the rated settings (Figure 1 and Figure 2). However, the extant literature did not permit hypotheses about BI group differences on the specific skills comprising SC, other than to assume that variations would reflect the extent to which BI impacts particular aspects of SC that are relevant to functioning in the observers’ settings. Parent-rated BI group differences in SC were found between children classified with High and Low BI on six scales (Assertion, Communication, Engagement, Self-Control, and Total Social Skills, and Internalizing Behaviors), four of which (italicized) remained significant after excluding children who were similarly classified by teachers. Teacher-rated BI group differences were found for four SC scales (Assertion, Cooperation, Engagement, and Internalizing Behaviors), two of which (italicized) remained significant after excluding children who were similarly classified by parents. Across informants, children with High BI were rated lower in Assertion and Engagement and had more Internalizing Behaviors than peers rated with Low BI. A central component of BI includes negative emotions, including fear and apprehension, which are associated with internalizing behaviors, and reluctance to participate, particularly in social, ambiguous, or novel situations [36]. Total SC was rated higher by teachers than by parents but there were fewer BI group differences in the number of teacher-rated skills across BI groupings. One possible explanation for this is that teachers of young children provide classroom structures and supports that may scaffold social, emotional, and academic competence [47] and may reduce the impact of BI on certain aspects of SC. Examining the link between BI and specific SCs may yield a more nuanced understanding of the functional relevance of BI for a child in each setting.*


This study is limited by its small sample size, particularly in the analyses of subsamples. Nevertheless, when children who were similarly classified by both informants were excluded, many of the group differences showing that children with higher BI were at a relative social disadvantage remained significant. Practitioners and researchers are rightly admonished to obtain information from multiple sources and informants [48]. Yet, differences in how these informants perceive the same child, often a fundamental source of misunderstanding, has been understudied. This study’s findings suggest that it may be counterproductive to seek informant agreement on a trait such as BI to understand its impact on functioning. Rather, children’s relative standing on BI reactivity in a particular setting should be considered in relation to functioning in that setting regardless of their standing on BI in other settings. However, the generalizability of findings is limited given that participants were largely middle-class and that all teachers were White. In developmental research, there is a call for greater focus on recruiting diverse samples to determine the generalizability of findings across sociocultural contexts [49].

School-based psychologists, social workers, and counselors can play a role in forging common ground and finding explanations for differences that honor both perspectives so that parents and teachers recognize and address concerns within each context. For example, parents and teachers may gain insight into the conditions in the milieu that elicit the child’s BI tendencies to ameliorate its disruptive impact on learning and functioning. This study’s finding that the inverse relation of BI with SC appears to be context/informant dependent also raises questions for future research. For example, should networks of associations among theoretically linked variables continue to be conceptualized as context free?

## Figures and Tables

**Figure 1 behavsci-14-01080-f001:**
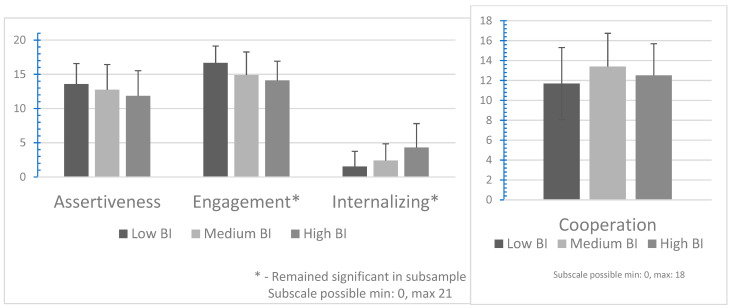
Mean Social Skills ratings for teachers.

**Figure 2 behavsci-14-01080-f002:**
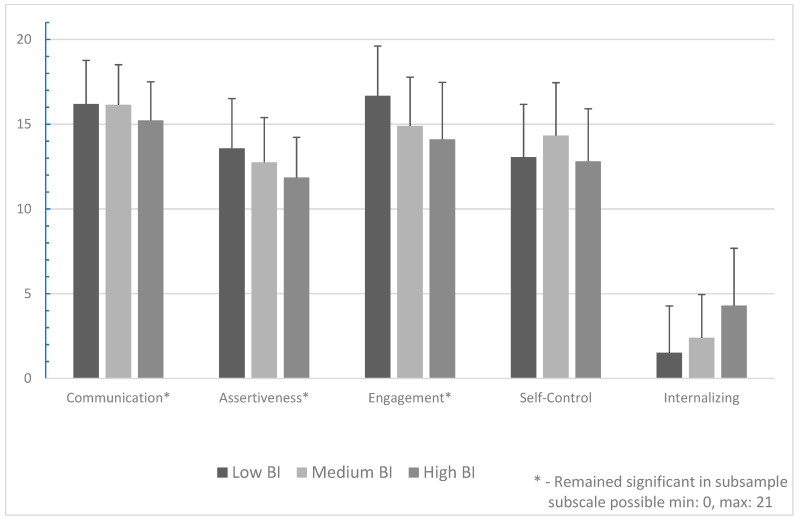
Mean Social Skills ratings for parents.

**Table 1 behavsci-14-01080-t001:** Descriptive data. Full sample.

Variable	BI Reactivity	N	M	SD	Range	N	M	SD	Range
		Teacher-Rated Social Skills				Parent-Rated Social Skills			
Social Skills Total Score	Low	50	101.22	12.95	74–126	41	101.29	12.09	80–128
	Medium	50	101.82	13.42	74–130	44	95.09	10.47	70–120
	High	51	98.47	11.14	67–116	43	89.44	10.9	59–115
	Total	151	100.49	12.54	67–130	128	95.18	12.07	59–128
Communication	Low	52	16.19	3.14	7–21	43	16.67	2.57	9–21
	Medium	53	16.15	3.07	9–21	44	15.68	2.36	11–19
	High	53	15.23	3.02	6–20	44	14.82	2.27	9–20
	Total	158	15.85	3.09	6–21	131	15.72	2.5	9–21
Cooperation	Low	52	11.69	3.62	5–18	43	12.84	2.35	7–18
	Medium	53	13.4	3.34	4–18	44	12.41	2.37	6–17
	High	53	12.51	3.19	6–18	44	12.41	2.62	7–17
	Total	158	12.54	3.43	4–18	131	12.55	2.44	6–18
Assertiveness	Low	52	13.58	2.99	5–21	43	15.7	2.93	6–20
	Medium	52	12.75	3.68	3–21	44	14.36	2.64	9–20
	High	53	11.85	3.68	0–17	44	13.14	2.38	8–18
	Total	157	12.72	3.52	0–21	131	14.39	2.84	6–20
Responsibility	Low	52	12.48	3.23	5–18	43	12.53	2.58	8–18
	Medium	52	13.37	3.12	6–18	44	11.68	2.41	3–17
	High	53	13.28	2.81	7–18	44	11.64	2.47	5–16
	Total	157	13.05	3.06	5–18	131	11.95	2.5	3–18
Empathy	Low	52	12.77	3.2	6–18	43	13.42	3.07	8–18
	Medium	53	12.15	3.31	5–18	44	12.14	3.08	3–18
	High	53	11.94	2.98	5–21	44	11.95	3.82	4–19
	Total	158	12.29	3.16	5–21	131	12.5	3.38	3–19
Engagement	Low	52	16.67	2.47	11–21	43	16.6	2.94	11–21
	Medium	52	14.9	3.37	5–21	44	14.98	2.87	7–21
	High	53	14.11	2.8	7–19	44	11.52	3.36	4–19
	Total	157	15.43	3.88	5–21	131	14.35	3.71	4–21
Self-Control	Low	52	13.06	4.06	2–20	43	11.86	3.11	5–18
	Medium	52	14.33	4.04	4–21	44	10.89	3.12	2–18
	High	53	12.81	3.95	1–20	44	10.11	3.1	3–16
	Total	157	13.61	4.85	1–47	131	10.95	3.17	2–18
Internalizing	Low	52	1.52	2.23	0–10	43	3.33	2.76	0–9
	Medium	53	2.4	2.45	0–9	44	3.77	2.55	0–10
	High	53	4.3	3.5	0–15	44	5.89	3.38	0–13
	Total	158	2.75	3	0–15	131	4.34	3.11	0–13

**Table 2 behavsci-14-01080-t002:** Descriptive data. Sub sample.

Variable	BI Reactivity	N	M	SD	Range	N	M	SD	Range
		Teacher-Rated Social Skills				Parent-Rated Social Skills			
Social Skills Total Score	Low	25	102.17	13.12	74–126	23	101.74	10.96	80–128
	Medium	24	100.38	12.61	77–130	27	97.41	10.97	77–120
	High	26	98.63	8.62	84–115	25	88.12	11.4	59–107
	Total	75	100.39	11.55	74–130	75	95.64	12.33	59–128
Communication	Low	25	16.48	3.39	7–21	24	16.75	2.63	9–21
	Medium	25	15.92	2.89	12–21	27	15.81	2.39	11–19
	High	26	15	2.21	11–20	25	14.12	1.94	9–18
	Total	76	15.79	2.89	7–21	76	15.55	2.54	9–21
Cooperation	Low	25	11.48	3.19	5–17	24	13.08	2.34	9–18
	Medium	25	13.44	3.58	5–18	27	12.48	2.72	6–17
	High	26	11.65	2.54	7–17	25	11.68	2.64	7–17
	Total	76	12.18	3.21	5–18	76	12.41	2.61	6–18
Assertiveness	Low	25	14.12	2.92	9–21	24	15.58	3.11	6–20
	Medium	25	12.71	3.84	7–21	27	14.96	2.41	9–20
	High	26	12.46	2.53	7–17	25	13.32	2.44	8–18
	Total	76	13.09	3.17	7–21	76	14.62	2.79	6–20
Responsibility	Low	25	12.72	3.61	5–18	24	12.58	2.5	8–18
	Medium	24	13.58	3.31	6–18	27	12.04	2.86	3–17
	High	26	12.5	2.25	9–18	25	10.96	2.56	5–16
	Total	75	12.92	3.09	5–18	76	11.86	2.7	3–18
Empathy	Low	25	13.24	2.7	7–17	24	13.17	2.81	8–18
	Medium	24	11.68	3.01	5–18	27	13	2.72	9–18
	High	26	12.19	2.67	9–18	25	11.48	3.98	4–18
	Total	75	12.37	2.83	5–18	76	12.55	3.26	4–18
Engagement	Low	25	17.08	2.12	12–21	24	16.58	2.45	13–21
	Medium	24	14.42	2.6	9–21	27	15.37	2.47	11–21
	High	26	14.38	2.61	7–19	25	12.48	3.32	6–19
	Total	75	15.29	2.74	7–21	76	14.8	3.22	6–21
Self-Control	Low	24	12.6	4.61	2–20	24	12.33	2.6	8–17
	Medium	24	13.88	4.33	4–21	27	11.26	3.59	2–18
	High	24	13.19	3.14	7–20	25	10.12	3.55	3–15
	Total	72	13.21	4.04	2–21	76	11.22	3.37	2–18
Internalizing	Low	25	1.4	2.2	0–9	24	3.46	2.87	0–9
	Medium	25	2.6	2.5	0–7	27	3.85	2.67	0–10
	High	26	5.08	4.04	0–15	25	4.68	3.13	0–12
	Total	76	3.05	3.37	0–15	76	4	2.9	0–12

Note: subsample includes children whose parents and teachers did NOT agree on BI level.

**Table 3 behavsci-14-01080-t003:** BI Reactivity and Social Skills ANOVAs full sample.

	SS	df	Mean Square	F	Sig.
**SSIS—Teacher Report**					
Assertion	78.43	154	39.22	3.26	0.04 *
Communication	31.5	155	15.75	1.66	0.19
Cooperation	76.27	155	38.14	3.33	0.04 *
Empathy	19.33	155	9.67	0.97	0.38
Engagement	179.92	154	89.96	10.68	0.00 **
Responsibility	24.9	154	12.45	1.33	0.27
Self-control	69.13	154	34.57	2.14	0.12
Social Skills Scale	323.07	148	161.54	1.03	0.36
Internalizing	213.04	155	106.52	13.73	0.00 **
**SSIS—Parent Report**					
Assertion	142.71	128	71.36	10.1	0.00 **
Communication	75.02	128	37.51	6.49	0.00 **
Cooperation	5.29	128	2.65	0.44	0.64
Empathy	55.19	128	27.6	2.47	0.09
Engagement	587.61	128	293.81	31.28	0.00 **
Responsibility	22.2	128	11.1	1.8	0.17
Self-control	66.6	128	33.3	3.45	0.04 *
Social Skills Scale	2948.14	125	1474.07	11.85	0.00 **
Internalizing	163.62	128	81.81	9.59	0.00 **

* Significant at the 0.05 level. ** Significant at the 0.01 level.

**Table 4 behavsci-14-01080-t004:** BI Reactivity and Social Skills ANOVAs sub sample.

	SS	df	Mean Square	F	Sig.
**SSIS—Teacher Report**					
Assertion	40.29	72	20.14	2.06	0.14
Communication	28.55	73	14.28	1.74	0.18
Cooperation	59.14	73	29.57	3.02	0.06
Empathy	31.65	73	15.82	2.03	0.14
Engagement	119.72	72	59.86	9.94	0.00 **
Responsibility	16.15	72	8.07	0.84	0.44
Self-control	19.92	72	9.96	0.6	0.55
Social Skills Scale	150.53	71	75.26	0.56	0.58
Internalizing	179.94	73	89.97	9.75	0.00 **
**SSIS—Parent Report**					
Assertion	67.7	73	33.85	4.79	0.01 *
Communication	87.58	73	43.79	8.05	0.00 **
Cooperation	24.34	73	12.17	1.83	0.17
Empathy	43.22	73	21.61	2.09	0.13
Engagement	219.67	73	109.84	14.31	0.00 **
Responsibility	33.65	73	16.83	2.39	0.1
Self-control	60.04	73	30.02	2.76	0.07
Social Skills Scale	2353.69	72	1176.84	9.53	0.00 **
Internalizing	19.19	73	9.6	1.15	0.32

Note: subsample includes children whose parents and teachers did NOT agree on BI level. * Significant at the 0.05 level. ** Significant at the 0.01 level.

## Data Availability

The raw data supporting the conclusions of this article will be made available by the authors on reasonable request.

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
