# Peer review of "Behavioral Inhibition and Social Competence Through the Eyes of Parent and Teacher Informants"

_behavsci, 2024, doi:10.3390/bs14111080_

Round 1
Reviewer 1 Report
Comments and Suggestions for Authors
This article posits a theory (Realistic Accuracy Model of Person Perception) to explain inter-informant discrepancy in BI ratings about children. This is an interesting and worthwhile research aim, but the writing needs to be improved for clarify and for highlighting the main contributions of the research. More specific comments follow:
The introduction is clear and well-written, but ideas can be a bit more elaborated upon to make it clearer how the three hypotheses on page 4 follow from the literature. The hypotheses themselves are not fully justified and it’s not clear why these specific hypotheses follow. Also, more examples similar to the example at the end of hypothesis 3 would be very helpful for the reader to understand exactly what the hypothesized patterns would look like in real life.
The RAM also needs to be a bit more clearly defined and explained the first time it is mentioned. Right now, there is text in the introduction about the implications of the RAM model for inter-informant discrepancy, but we the reader doesn't have a good sense of the model itself is and what specific framework is represents.
Additionally, although I think the analysis methods are appropriate, the analyses are not clearly communicated. For example, can the authors make figures that clearly communicate the main findings? I had a hard time figuring out which findings are central to the argument here. Is it that there is discrepancy confirming prior work? Is it that despite inter-informant discrepancy, there are significant associations between BI and various SC capabilities? Relatedly, the main take-away of the work is lost on me. What is novel and unique about this work and how do the findings inform the field in theory and practice?
I recommend the authors streamline and clarify the finding, thinking carefully about the main point of the work and figuring out how to best highlight and bring that point to the foreground through the analysis and data visualization.
Reviewer 2 Report
Comments and Suggestions for Authors
Interesting study. However, it can be improved and even deepened in future research.
Lines 132-133: “Kagan and colleagues (Kagan et al., 1988) estimate that BI…”
1. This is not a correct presentation of the reference for this journal.
2. Hypothesis 1 presents more than one objective, split into two sentences. Please correct that to avoid misunderstandings about what the first hypothesis is.
3. In practice, you have 125 cases. However, you present n=176 in the abstract. Of course, it is the number of parents and teachers, but the study concerns the children's assessment.
4. I presume that you assume that the measurement scales are reliable and valid for this sample. However, it would be worth saying something about that because the psychometric characteristics of scales can change among different samples.
5. You describe children’s sex and ethnicity. Were there any differences between these variable categories?
6. All teachers were identified as White and Female. Do you think this could have any particular impact on the results? Could this be a limitation, an advantage, or both in the study?
7. What did you do when you had the two parents responding? All the children had two parents?
8. It would probably be better to present more tables in the results section rather than describe all those figures in the text. The text could highlight what is essential in the findings.
9. The presentation of the results, discussion, and conclusion would have been better if you had mentioned the numbered hypotheses.
Round 2
Reviewer 1 Report
Comments and Suggestions for Authors
This is a very responsive revision and the writing is much clearer now. The work is interesting and applicable. I have a few more suggestions to improve data presentation and the discussion of patterns:
· Tables are still very hard to parse out. I recommend making the row heights much smaller and fitting each table on one page. Also the ANOVA table (table 3) needs to specify what the second column numbers are.
· Figures: What is the possible maximum average for each of the subscales of the SSIS-RS scale? Do they all have different possible ranges? If so, showing these on the same figure can be confusing. For example, if the possible maximum score for “cooperation” is different from the maximum score for “Communication”, then Figures 1 and 2 can be misleading. The y axis needs to show the full possible range for the subscales, which means that if they have the same range, they can all be shown on one; if not, I recommend using the “facets” argument in ggplot in R to show each subscale in different facet of the plot (each facet can have a different y range).
· Figures should also show SDs or SEs.
· In the Discussion section, it would be a good idea to speculate about why parent-reported BI did not correlate with any teacher-reported SC subscales, but teacher-reported BI correlated with three parent-reported SC scales.
